# TimeHC-RL: Temporal-aware Hierarchical Cognitive Reinforcement Learning for Enhancing LLMs' Social Intelligence

## Abstract

Recently, Large Language Models (LLMs) have made significant progress in IQ-related domains that require careful thinking, such as mathematics and coding. However, enhancing LLMs' cognitive development in social domains, particularly from a post-training perspective, remains underexplored. Recognizing that the social world follows a distinct timeline and requires a richer blend of cognitive modes (from intuitive reactions (System 1) and surface-level thinking to deliberate thinking (System 2)) than mathematics, which primarily relies on System 2 cognition (careful, step-by-step reasoning), we introduce **T**emporal-aware **H**ierarchical **C**ognitive **R**einforcement **L**earning **(TimeHC-RL)** for enhancing LLMs' social intelligence. In our experiments, we systematically explore improving LLMs' social intelligence and validate the effectiveness of the TimeHC-RL method, through five other post-training paradigms and two test-time intervention paradigms on eight datasets with diverse data patterns. Experimental results reveal the superiority of our proposed TimeHC-RL method compared to the widely adopted System 2 RL method. It gives the 7B backbone model wings, enabling it to rival the performance of advanced models like DeepSeek-R1 and OpenAI-O3. Additionally, the systematic exploration from post-training and test-time interventions perspectives to improve LLMs' social intelligence has uncovered several valuable insights.

## 1 Introduction

Recently, Large Language Models (LLMs) have made significant progress and achieved notable success in IQ-related domains such as mathematics and coding. Several approaches have contributed to this progress: Long-thought Supervised Fine-Tuning (SFT) (even with relatively small data scales, as seen in LIMO (Ye et al., 2025)), rule-based Reinforcement Learning (RL) (as demonstrated by DeepSeek-R1 (Guo et al., 2025) and OpenAI-O1 (Jaech et al., 2024)), and test-time budget forcing (as implemented in s1 (Muennighoff et al., 2025)). However, advancing the cognitive development of LLMs in social domains, despite its significance, has not received sufficient attention and comprehensive exploration.

Current approaches to enhance LLMs' cognitive performance in the social domain can mainly be divided into three categories: (1) Prompt-based approaches, such as perspective-taking (Wilf et al., 2024; Jung et al., 2024); (2) External tool-based approaches, such as building world models (Huang et al., 2024a), belief trackers (Sclar et al., 2023), and solvers (Hou et al., 2024b); (3) Model-based approaches, such as Bayesian models (Zhang et al., 2025; Shi et al., 2025; Jin et al., 2024). Despite these advances, **there remains a notable research gap in systematic exploration from post-training and test-time intervention perspectives.**

We first conduct a evaluation and analysis of the advanced DeepSeek-R1 model's performance on social domain benchmark, specifically ToMBench (Chen et al., 2024) and HiToM (Wu et al., 2023). Correctly answering questions in ToMBench requires advanced cognition and understanding of social contexts, while correctly answering questions in HiToM requires sophisticated reasoning about interpersonal dynamics in social-event lines. Our experiments show that DeepSeek-R1 consumes a large number of tokens on both benchmarks; its performance on ToMBench is on par with the GPT-4 series models (78.4% vs 75.3%, as seen in Appendix A.1), while its results on HiToM are noticeably

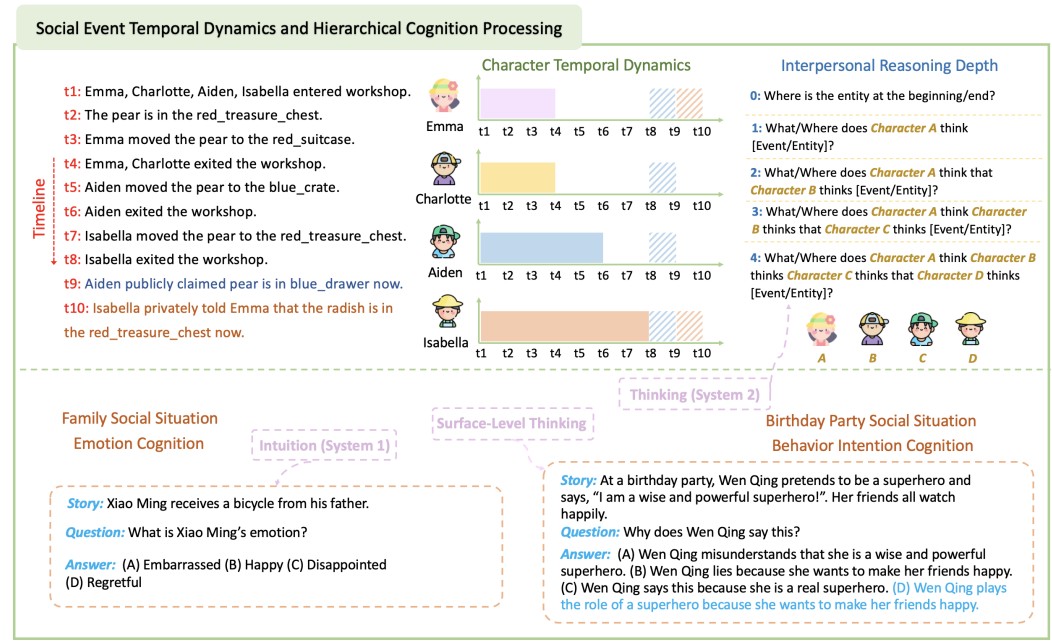

Figure 1: Top: Real-world social events following a clear timeline and character temporal dynamics, interpersonal reasoning presentation. Bottom: Presentation of social situation cognition; Diverse cognitive patterns observed in the social domain.

stronger. Based on experimental observations, we analyze the adaptability of DeepSeek-R1's training paradigm in the social domain:

- Its good performance on HiToM benefits from its superior reasoning capabilities, which are incentivized by rule-based RL. Additionally, the social events in HiToM are relatively basic and simple, requiring lower levels of social situation cognition.

- Its relatively average performance on ToMBench may be due to a lack of diverse social situations in its training data.

- It consumes a large number of tokens on ToMbench. Yet, **unlike the mathematics domain, where System 2 cognition (careful, step-by-step reasoning) is predominant, the social domain involves a richer mix of cognitive modes**: cognition of social situations can be intuitive (System 1) (Kahneman, 2011) or involve some basic analytical understanding, while inferring others' mental states may require more deliberate thinking (System 2).

Building upon this analysis, we implement two RL paradigms: one where the model answers directly, and another where it first thinks before answering. Surprisingly, **the model perform better on ToMbench under the direct-answer RL paradigm compared to the think-then-answer one** (79.2% vs 76.9%). This further reinforces our analysis of the diversity of cognitive patterns in the social domain. In light of these, we introduce **T**emporal-aware **H**ierarchical **C**ognitive **R**einforcement **L**earning (**TimeHC-RL**) for Enhancing LLMs' Social Intelligence. Our key methodological contributions include:

- **Addressing real-world temporal dynamics**: Social events and conversations inherently follow temporal sequences with distinct characteristics (Figure 1, Top). Conventional rewards focused merely on format and outcomes prove inadequate for training LLMs to reason effectively across social event timelines and conversation flows. To address this limitation, we introduce a temporal-aware advantage-level reward mechanism.

- **Implementing hierarchical cognitive processing**: In response to the diverse cognitive patterns observed in the social domain (Figure 1, Bottom), we propose a hierarchical cognition framework that encompasses a spectrum from intuitive reactions (System 1) and surface-level thinking to deliberate thinking (System 2).

Table 1: ToMi and Hi-ToM's social-event lines are mainly limited to simple object location changes and characters entering and exiting rooms; ExploreToM expands the action space, introducing richer interaction types such as communication between characters and secret observations; while other data sources contain situations, event lines, and conversation flows that are highly aligned with real-world social interactions. Based on the varying degrees of real-world cognitive demands these data sources require, we categorize them into three levels: Basic, Intermediate, and Advanced.

| Data Source | Real World Cognition Demand | Interpersonal Reasoning Depth | Usage |
|---|---|---|---|
| ToMi | Basic | 2 | Split for Post-Training (800) and In-Domain Evaluation (200) |
| Hi-ToM | Basic | 4 | Reasoning Depth 1,2 for Post-Training (360) Reasoning Depth 3,4 for In-Domain Evaluation (240) |
| ExploreToM | Intermediate | 2 | Split for Post-Training (2k) and In-Domain Evaluation (300) |
| ToMBench | Advanced | 1 | Split for Post-Training (2.4k) and In-Domain Evaluation (431) |
| SocialIQA | Advanced | 1 | Split for Post-Training (2k) and In-Domain Evaluation (120) |
| SimpleToM | Advanced | 1 | Out-of-Domain Evaluation (120) |
| OpenToM | Advanced | 2 | Out-of-Domain Evaluation (85) |
| ToMATO | Advanced | 2 | Out-of-Domain Evaluation (50) |

In our experiments, we systematically explore improving LLMs' social intelligence and validate the effectiveness of the TimeHC-RL method, through five other post-training paradigms and two test-time intervention paradigms on eight datasets with various data patterns. We use ToMi (Le et al., 2019), HiToM (Wu et al., 2023), ExploreToM (Sclar et al., 2024), ToMBench (Chen et al., 2024), SocialIQA (Sap et al., 2019)) as training sets to cultivate LLMs' comprehensive abilities in social situation cognition and sophisticated reasoning about interpersonal dynamics, while using SimpleToM (Gu et al., 2024), ToMATO (Shinoda et al., 2025), OpenToM (Xu et al., 2024)) for Out-Of-Distribution (OOD) evaluation. Our proposed TimeHC-RL method gives the 7B backbone model wings, enabling it to rival the performance of advanced models like DeepSeek-R1 and OpenAI-O3. Systematic exploration from post-training and test-time interventions perspectives to improve LLMs' social intelligence reveal: **(1)** SFT memorizes and has limited memory capacity, while RL generalizes. **(2)** RL implements more effective interpersonal reasoning depth extrapolation. **(3)** Cognition of social situations cannot be developed through test-time sequential scaling. **(4)** RL with different cognitive modes shows significant preferences in performance on different types of data.

## 2 DATASET CONSTRUCTION

Real-world social situations are remarkably diverse, and reasoning chains to infer others' mental states can be deeply layered, for example, questions like 'Where does Bob think Alice thinks...?' (Second-Order Mental Inference, Interpersonal Reasoning Depth 2). To enhance the social capabilities of LLMs, it is crucial to construct dataset that embodies a wide variety of social situations and necessitates sophisticated reasoning of interpersonal dynamics. In this section, we will introduce the dataset we build for LLM Post-Training and Evaluation, while highlighting several exceptional designs. Table 1 presents, for each data source, the depth of interpersonal reasoning required by its questions, the extent of real-world cognition demanded, and how the data are utilized for training and evaluation. The specific forms of samples in each data source can be found in Appendix A.6.

### 2.1 DATASET FOR LLM POST-TRAINING AND EVALUATION

**Post-Training.** We combine data from ToMi (Le et al., 2019), Hi-ToM (Wu et al., 2023), Explore-ToM (Sclar et al., 2024), ToMBench (Chen et al., 2024), and SocialIQA (Sap et al., 2019) to form our Post-Training dataset. The data format for ToMi, Hi-ToM, and ExploreToM is (Social-Event Lines, Question, Choices), which requires sophisticated reasoning about the interpersonal dynamics in the Social-Event Lines to answer the question correctly. The data format for ToMBench and SocialIQA is mainly (Social Situation, Question, Choices), which requires advanced cognition and understanding of the social situation to answer the question correctly.

**Evaluation.**   We divide the evaluation into In-Domain evaluation and OOD evaluation. For the In-Domain evaluation, we select non-overlapping data from ToMi, ExploreToM, ToMBench, and SocialIQA that were not used during the Post-Training phase, and Hi-ToM data with interpersonal reasoning depths of 3 and 4. For the OOD evaluation, we select data from SimpleToM (Gu et al., 2024), OpenToM (Xu et al., 2024), and ToMATO (Shinoda et al., 2025). The ToMATO dataset consists of dialogues between two agents generated with the Sotopia (Zhou et al., 2023) framework; its data format is (Conversation, Question, Choices). OpenToM adopts the same data format as ToMi, Hi-ToM, and ExploreToM, whereas SimpleToM shares the data format used by ToMbench and SocialIQA. The exceptional designs in our evaluation are summarized below:

- **Interpersonal reasoning depth generalization** – In Hi-ToM, samples with reasoning depths 1 and 2 are used for post-training, while depths 3 and 4 are reserved for in-domain evaluation.

- **Inference of agents' mental states during dialogue interaction** - Using Sotopia-generated agent-agent dialogues, evaluating the ability to infer agents' mental states during the dialogue interaction. (ToMATO)

- **Beyond social-situation cognition** – Not only assess social-situation cognition but also behavior prediction and judgment. (SimpleToM)

## 3 METHODS

### 3.1 PRELIMINARY

**Group Relative Policy Optimization**   Unlike SFT, which optimizes models through token-level losses, RL-based methods like GRPO utilize policy gradients, calculated from the reward loss, for optimization (Li et al., 2025b). This encourages exploring a much larger solution space (Guo et al., 2025).

Let $Q$ be the question set, $\pi_{\theta_{\text{old}}}$ be the policy model and $\{o_1, o_2, \cdots, o_G\}$ be a group of responses from $\pi_{\theta_{\text{old}}}$ for a question $q$. Let $\pi_{\theta_{\text{ref}}}$ denote the frozen reference model. The GRPO algorithms aim to optimize model $\pi_\theta$ by the following objective:

$$J_{\text{GRPO}}(\theta) = \mathbb{E}_{q \sim Q, \{o_i\}_{i=1}^G \sim \pi_{\theta_{\text{old}}}}$$

$$\left[ \frac{1}{G} \sum_{i=1}^{G} \min \left( \frac{\pi_\theta(o_i|q)}{\pi_{\theta_{\text{old}}}(o_i|q)} A_i, \text{clip} \left( \frac{\pi_\theta(o_i|q)}{\pi_{\theta_{\text{old}}}(o_i|q)}, 1 - \epsilon, 1 + \epsilon \right) A_i \right) - \beta D_{\text{KL}}(\pi_\theta \| \pi_{\text{ref}}) \right],$$

where $\epsilon$ and $\beta$ are clipping hyper-parameter and the coefficient controlling the Kullback–Leibler (KL) penalty, respectively. Here, $A_i = \frac{r_i - \text{mean}(\{r_1, r_2, ..., r_G\})}{\text{std}(\{r_1, r_2, ..., r_G\})}$ is the advantage using the group reward $\{r_1, r_2, \ldots, r_G\}$, and $D_{KL}(\pi_\theta \| \pi_{\text{ref}}) = \frac{\pi_{\text{ref}}(o_i|q)}{\pi_\theta(o_i|q)} - \log \left( \frac{\pi_{\text{ref}}(o_i|q)}{\pi_\theta(o_i|q)} \right) - 1$ is the KL divergence loss. GRPO eliminates the critic model in PPO by estimating the relative advantage by sampling a group of responses $\{o_i\}_{i=1}^G$ and normalizing their rewards within the group to compute a relative advantage, which is more computationally efficient (Shao et al., 2024).

**Dual-System Theory: Two Modes of Cognitive Processing**   The Dual-System Theory (Sloman, 1996; Kahneman, 2011; Evans & Stanovich, 2013) offers a framework for understanding human cognition. System 1 is characterized by its rapidity, strong intuitive nature, and effortlessness; it handles vast amounts of information in daily life and produces immediate responses. In contrast, System 2 is slow, analytical, and requires conscious attentional investment, a deliberate thinking process that plays a crucial role in complex problem solving.

### 3.2 TEMPORAL-AWARE HIERARCHICAL COGNITIVE REINFORCEMENT LEARNING

#### 3.2.1 HIERARCHICAL COGNITION FRAMEWORK

Cognition of social situations can be intuitive or involve some basic analytical understanding, while inferring others' mental states may require more deliberate thinking. In light of the diverse cognitive

patterns observed in the social domain, we propose a hierarchical cognition framework that ranges from intuitive reaction (System 1), surface-level thinking, to deliberate thinking (System 2). The corresponding behavioral tags for these cognition modes are:

- **System1**: <answer> final answer </answer>.
- **Surface-level Thinking**: <social context understanding> ... </social context understanding> + <answer> final answer </answer>.
- **System2**: <think> thought process </think> + <answer> final answer </answer>.

These tags are formulated in alignment with linguistic principles.

### 3.2.2 TRAINING TEMPLATE

Considering the distinctive characteristics of data in the social domain, we employ a simple and intuitive prompt to guide the model to adaptively select one of the three cognitive modes under the hierarchical cognition framework, with the complete training template provided in Figure 2.

> You are a helpful assistant. Now the user asks you to solve a problem. Adaptively choose one of the following three cognitive modes to solve the problem: (1) Based on intuition, directly output the answer in <answer></answer> tags. (2) Perform understanding and analysis of the social context, then provide the user with the answer. Please output the social context understanding and analysis in <social context understanding> </social context understanding> and final answer in <answer> </answer> tags. (3) Carefully think about the problem, then provides the user with the answer. Please output the thinking process in <think> </think> and final answer in <answer> </answer> tags.

Figure 2: Training template for a model to adaptively choose among three cognitive modes: intuition, surface-level thinking, and deliberate thinking.

### 3.2.3 REWARD MODELING

The reward function consists of three components: format reward, outcome reward, and temporal advantage-level reward. The format reward validates that responses adhere to the required structural format, ensuring all elements appear in the correct sequence and are enclosed within appropriate tags:

$$r_{\text{format}} = \begin{cases} 1, & \text{if format is correct} \\ -1, & \text{if format is incorrect} \end{cases}$$

The outcome reward is a rule-based metric that verifies whether the content enclosed in the <answer> </answer> tags exactly matches the ground truth (gt) label, which is designed as follows:

$$r_{\text{accuracy}} = \begin{cases} 2, & \text{if answer tag exists and extracted answer matches gt label} \\ -1.5, & \text{otherwise} \end{cases}$$

The temporal advantage-level reward is a contrastive reward mechanism that explicitly encourages the construction of temporal logic flows. The core idea involves comparing the model's performance on the same social question when social-event lines or conversation flow are provided in two different orders: (1) the temporally ordered sequences, and (2) a shuffled version. For each input question, we generate two groups of responses $\{o_i\}_{i=1}^{G}$ and $\{\tilde{o}_i\}_{i=1}^{\tilde{G}}$ using the ordered and shuffled inputs, respectively. Let $p$ and $\tilde{p}$ denote the proportion of correct answers in each group. We then define a temporal advantage-level reward as:

$$A_{temporal} = \begin{cases} \alpha, & \text{if } p > \mu \cdot \tilde{p} \\ 0, & \text{otherwise} \end{cases}$$

where $\alpha$ and $\mu$ are hyper-parameters. Here we set $\alpha = 0.1$, $\mu = 0.8$. As for hyperparameter $\mu$, although intuitively setting it to 1.0 would seem more reasonable, our experimental results show that

moderate relaxation can lead to better model performance. This represents a trade-off between noisy signals and obtaining denser learning signals. Additionally, due to random sampling, the performance of shuffle groups exhibits uncertainty. $\mu$ is a balanced choice under the interaction of multiple factors.

This contrastive design incentivizes the model to perform better when the social-event lines or conversation flow is presented in correct temporal order than when it is shuffled. The model receives this positive reinforcement only when its response strategy for a specific question demonstrates clear dependence on temporal information. The $A_{temporal}$ is selectively applied only to correct responses, when the model successfully leverages temporal patterns, correct responses receive enhanced reinforcement through this higher reward, while incorrect responses remain unaffected.

# 4 EXPERIMENTS

## 4.1 VARIOUS PARADIGMS SETUP USED FOR SYSTEMATIC EXPLORATION AND AS BASELINES

### 4.1.1 MULTIPLE POST-TRAINING PARADIGMS

To verify the effectiveness of the TimeHC-RL method and systematically explore improving LLMs' social intelligence from a post-training perspective, we implement multiple post-training paradigms.

**Direct SFT.** Using social-event lines / social situations and questions as input, with answers as output, to conduct direct SFT on the model.

**Long-thought SFT.** Using the DeepSeek-R1-Distill-Qwen-32B model (Guo et al., 2025), we generate detailed chains of thought for each training sample. After applying basic rule-based filtering to remove low-quality and inconsistent false outputs, we obtain a high-quality long-thought dataset, which is employed for subsequent SFT.

**RL with System 1.** Unlike RL with System 2 cognition, which encourages models to think deliberately, RL with System 1 cognition prompts models to generate answers intuitively and directly: "Please directly output the answer based on intuition." RL with System 1 cognition eliminates the format reward and retains only the outcome reward.

**RL with System 2.** Following Guo et al. (2025)'s paradigm, we design a prompt that encourages models to engage in a deliberate thinking process before producing the final answer. The prompt is defined as follows: "Please output the deliberate thinking process in <think> </think> and final answer in <answer> </answer> tags, respectively." The reward function consists of two components: format reward and outcome reward.

**HC-RL.** Reinforcement learning based on the hierarchical cognitive framework and training template proposed in Section 3.2.1 and 3.2.2.

### 4.1.2 TEST-TIME INTERVENTION: PARALLEL SCALING AND SEQUENTIAL SCALING

In addition to the post-training paradigm, we also explore improving LLMs' social intelligence from the perspective of Test-Time Intervention: Parallel Scaling and Sequential Scaling.

**Parallel Scaling: Majority Voting.** Repeatedly sample N candidates from the model for each input. From these candidates, select the one that appears most frequently as the final answer (Snell et al., 2024).

**Sequential Scaling: Budget Forcing.** To let the model spend more test-time computing on a problem, when the model is about to complete its solution to a problem, append "Wait" to the model's current reasoning trace to encourage the model to engage in more thinking and exploration (Muennighoff et al., 2025).

Table 2: Performance evaluation of multiple methods and advanced foundation models in In-Domain scenarios (The HiToM (Third) and HiToM (Fourth) columns also include generalization assessment of interpersonal reasoning depth, because in the post-training phase, we only use interpersonal reasoning problems with reasoning depths of 1 and 2 from HiToM). $\Delta_{\text{Backbone}}$ represents the performance improvement brought by our proposed TimeHC-RL method when applied to the backbone model, and $\Delta_{\text{RL with System 2}}$ represents the performance advantage of our proposed TimeHC-RL method compared to the widely adopted system 2 RL paradigm. Due to space constraints, a more comprehensive performance comparison of our method and existing mainstream LLMs can be found in the Appendix A.7.

| Model | ToMi | ExploreToM | ToMBench | SocialIQA | HiToM (Third) | HiToM (Fourth) | AVG |
|---|---|---|---|---|---|---|---|
| BackBone Models | | | | | | | |
| Qwen2.5-7B-Instruct-1M | 0.60 | 0.45 | 0.69 | 0.77 | 0.29 | 0.26 | 0.51 |
| Advanced Foundation Models | | | | | | | |
| GPT-4o | 0.74 | 0.57 | 0.80 | 0.84 | 0.35 | 0.32 | 0.60 |
| DeepSeek-R1 | 0.93 | 0.79 | 0.78 | 0.83 | 0.70 | 0.69 | 0.79 |
| OpenAI-O3 | 0.91 | 0.82 | 0.84 | 0.86 | 0.72 | 0.70 | 0.81 |
| Our Implemented Models | | | | | | | |
| Direct SFT | 0.72 | 0.53 | 0.39 | 0.25 | 0.56 | 0.50 | 0.49 |
| Long-thought SFT | 0.89 | 0.74 | 0.73 | 0.63 | 0.55 | 0.42 | 0.66 |
| RL with System 1 | 0.84 | 0.93 | 0.79 | 0.81 | 0.60 | 0.53 | 0.75 |
| RL with System 2 | 0.90 | 0.94 | 0.77 | 0.78 | 0.67 | 0.60 | 0.78 |
| HC-RL | 0.92 | 0.94 | 0.81 | 0.79 | 0.65 | 0.64 | 0.79 |
| TimeHC-RL | 0.96 | 0.95 | 0.82 | 0.78 | 0.68 | 0.64 | 0.81 |
| $\Delta_{\text{Backbone}}$ | +0.36 | + 0.50 | + 0.13 | + 0.01 | +0.39 | + 0.38 | +0.30 |
| $\Delta_{\text{RL with System 2}}$ | +0.06 | +0.01 | + 0.05 | + 0.00 | + 0.01 | + 0.04 | +0.03 |
| Human Performance | | | | | | | |
| Human | - | - | 0.85 | 0.84 | - | - | - |

## 4.2 IMPLEMENTATION DETAILS

We use the Qwen2.5-7B-Instruct-1M model (Yang et al., 2024) as the backbone model, which has performance comparable to the Qwen2.5-7B-Instruct model and can handle longer context. We use the TRL (von Werra et al., 2020) and VeRL (Sheng et al., 2024) frameworks to implement SFT-based methods and RL-based methods, respectively. The specific parameter configurations used for SFT and RL can be found in the Appendix A.3. We conduct all experiments on 8 A100 (80GB) GPUs. In addition, we evaluate the performance of the advanced foundation models GPT-4o[1], DeepSeek-R1 (Guo et al., 2025), and OpenAI-O3[2] as references.

## 4.3 METRICS AND RELIABLE REWARD SIGNAL

We use the accuracy of question answering as a metric to measure performance. To ensure the reliability and accuracy of the reward signal for stable and effective RL training, we have conducted many detailed data processing steps. For example, in ToMBench, when the answer is given as an option name like "A", we match it with the corresponding specific answer content. Or when the model's response is in the format "A. answer", although it doesn't strictly match the answer, we still consider it correct.

## 4.4 MAIN RESULTS

As demonstrated in Table 2, our TimeHC-RL method delivers a substantial 30.0 points comprehensive performance improvement over the backbone model in the In-Domain evaluation. It also surpasses the widely adopted System 2 RL paradigm by 3.0 points. Remarkably, with just 7B parameters, our method achieves a comprehensive performance score of 81.0%, comparable to state-of-the-art models like DeepSeek-R1 and OpenAI-O3. The Direct SFT method and Long-thought SFT method achieve comprehensive performances of 49.0% and 66.0%, respectively, showing a notable gap compared to the RL paradigm.

---

[1] https://openai.com/index/hello-gpt-4o/
[2] https://openai.com/index/introducing-o3-and-o4-mini/

Table 3: Performance evaluation of multiple methods and advanced foundation models in OOD scenarios. $\Delta_{\text{Backbone}}$ and $\Delta_{\text{RL with System 2}}$ represents the same meaning as explained in Table 2 caption. Due to space constraints, a more comprehensive performance comparison of our method and existing mainstream LLMs can be found in the Appendix A.7.

| Model | ToMATO (First) | ToMATO (Second) | SimpleToM (Behavior) | OpenToM (Attitude) | OpenToM (Location) | AVG |
|---|---|---|---|---|---|---|
| BackBone Models | | | | | | |
| Qwen2.5-7B-Instruct-1M | 0.72 | 0.68 | 0.17 | 0.56 | 0.61 | 0.55 |
| Advanced Foundation Models | | | | | | |
| GPT-4o | 0.84 | 0.92 | 0.13 | 0.68 | 0.81 | 0.68 |
| DeepSeek-R1 | 0.80 | 0.80 | 0.60 | 0.76 | 0.84 | 0.76 |
| OpenAI-O3 | 0.88 | 0.96 | 0.25 | 0.88 | 0.86 | 0.77 |
| Our Implemented Models | | | | | | |
| Direct SFT | 0.12 | 0.24 | 0.17 | 0.03 | 0.03 | 0.12 |
| Long-thought SFT | 0.40 | 0.48 | 0.32 | 0.60 | 0.69 | 0.50 |
| RL with System 1 | 0.64 | 0.64 | 0.22 | 0.60 | 0.68 | 0.56 |
| RL with System 2 | 0.68 | 0.72 | 0.27 | 0.56 | 0.69 | 0.58 |
| HC-RL | 0.72 | 0.76 | 0.30 | 0.64 | 0.70 | 0.62 |
| TimeHC-RL | 0.80 | 0.80 | 0.35 | 0.60 | 0.73 | 0.66 |
| $\Delta_{\text{Backbone}}$ | +0.08 | +0.12 | +0.18 | +0.04 | +0.12 | +0.11 |
| $\Delta_{\text{RL with System 2}}$ | +0.12 | +0.08 | +0.08 | +0.04 | +0.04 | +0.08 |
| Human Performance | | | | | | |
| Human | 0.87 | - | - | 0.86 | - | - |

As demonstrated in Table 3, in the OOD evaluation, we find that during the Post-training phase, our proposed TimeHC-RL method for cultivating social situation cognition and interpersonal reasoning abilities in LLMs brings a 11.0 points improvement, outperforming the System 2 RL paradigm 8.0 points. Meanwhile, SFT-based methods, whether Direct SFT or Long-thought SFT, both reduce the original performance of the backbone model.

Comparing the performance of HC-RL and TimeHC-RL methods in both In-Domain and OOD evaluation scenarios, we find that the introduction of temporal rewards brings performance advantages of 2.0 points and 4.0 points, respectively. This indicates that it can integrate well with the hierarchical cognitive framework, collaboratively enhancing the social intelligence of LLMs.

## 4.5 IN-DEPTH ANALYSIS

**SFT memorizes, and has limited memory capacity (Direct SFT), while RL generalizes.** Both direct SFT and long-thought SFT reduce the original performance of the backbone model in OOD evaluation. In contrast, the RL paradigm still provides more or less gains to the backbone model. Furthermore, as shown in Table 2, direct SFT performs relatively well on ToMi, ExploreToM, HiToM (Third), and HiToM (Fourth), which are datasets focusing on interpersonal reasoning, but performs poorly on ToMBench and SocialIQA, which focus on social situation cognition. This indicates that the direct SFT method has lower tolerance for data patterns, which is very unfavorable for the de-

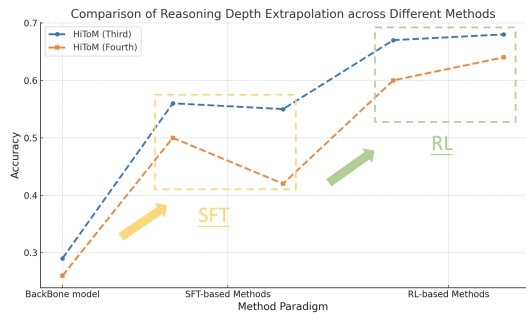

Figure 3: Performance comparison of SFT-based and RL-based methods on interpersonal reasoning depth extrapolation.

velopment of social intelligence, considering the inherent complexity of social intelligence. However, the long-thought SFT method improves this aspect, demonstrating better memory capacity.

**RL implements more effective interpersonal reasoning depth extrapolation.** Compared to direct SFT and long-thought SFT methods, although all methods underwent post-training on interpersonal reasoning problems with reasoning depths of 1 and 2, the RL method significantly outperforms on interpersonal reasoning problems with reasoning depths of 3 and 4 (0.68, 0.64), substantially surpassing both direct SFT (0.56, 0.50) and long-thought SFT (0.55, 0.42) methods, as shown in Figure 3.

Table 4: Performance evaluation of applying majority voting strategy to the Qwen2.5-7B-Instruct-1M backbone model, as well as applying budget forcing to the Long-thought SFT model.

| Model | ToMi | HiToM (Third) | HiToM(Fourth) | ToMBench (Moral Emotion) | ToMATO | AVG |
|---|---|---|---|---|---|---|
| Majority Voting Parallel Scaling | | | | | | |
| Qwen2.5-7B-Instruct-1M | 0.60 | 0.29 | 0.26 | 0.65 | 0.68 | 0.50 |
| Majority Voting (N = 8) | 0.71 | 0.22 | 0.20 | 0.75 | 0.56 | 0.49 |
| Δ | +0.11 | -0.07 | -0.06 | +0.10 | -0.12 | -0.01 |
| Budget Forcing Sequential Scaling | | | | | | |
| Long-thought SFT | 0.89 | 0.55 | 0.42 | 0.70 | 0.40 | 0.60 |
| Budget Forcing (M = 1) | 0.90 | 0.58 | 0.49 | 0.68 | 0.44 | 0.62 |
| Δ | +0.01 | +0.03 | +0.07 | -0.02 | +0.04 | +0.02 |

**Cognition of social situations cannot be developed through test-time sequential scaling.** As shown in Table 4, we explore the utility of Majority Voting (Parallel Scaling) and Budget Forcing (Sequential Scaling) in the social domain. We find that the application of the majority voting strategy did not demonstrate any clearly capturable characteristics, whether focusing on advanced cognition of social situations or interpersonal reasoning data. Budget forcing shows gains for data focusing on interpersonal reasoning, but had no effect on data focusing on advanced cognition of social situations. We speculate that developing social situational cognition may necessarily require either incorporating diverse social situations in the training data or increasing the model size.

**RL with different cognitive modes shows significant preferences in performance on different types of data.** As shown in Table 2, careful observation of the performance of RL with System 2 versus RL with System 1 on ToMi, ExploreToM, ToMBench, and SocialIQA reveals that RL with System 2 performs better on ToMi and ExploreToM datasets that focus on interpersonal reasoning, while RL with System 1 performs better on ToMBench and SocialIQA datasets that focus on social situational cognition. This further demonstrates the necessity of building a hierarchical cognitive framework to develop social intelligence in LLM. In Appendix A.5, we present how LLM adaptively employs different cognitive modes to address various data types in the social domain.

## 5 CONCLUSIONS, LIMITATIONS AND FUTURE WORKS

In this paper, considering the temporal dynamics of real-world social events and that the social domain involves a richer mix of cognitive modes (from intuitive reaction, surface-level thinking, to deliberate thinking), we introduce **T**emporal-aware **H**ierarchical **C**ognitive **R**einforcement **L**earning (**TimeHC-RL**) to enhance LLMs' social intelligence. We obtain a 7B parameter model that demonstrates strong comprehensive capabilities in social situation cognition and interpersonal reasoning, performing well across multiple social domain benchmarks. In our experiments, we systematically explore improving LLMs' social intelligence and validate the effectiveness of the TimeHC-RL method, through five other post-training paradigms and two test-time intervention paradigms on eight datasets with diverse data patterns, revealing several valuable insights that lay the foundation for future research on social intelligence in LLMs. Some limitations and potential future works are listed as follows:

- **Beyond Situational Intelligence and Cognitive Intelligence** In our paper, we focus more on situational intelligence and cognitive intelligence. The ability to behave and interact (behavioral intelligence) is also very important.

- **Scalable Social Situation Framework** We believe that incorporating richer social situations in training data, exposing LLMs to a more diverse social world, is very helpful for enhancing the social intelligence of LLMs. Therefore, forming a scalable social situation framework is extremely important.

- **Experiments with multiple model sizes** In our paper, we only conduct experiments with a 7B model. Considering that models of different sizes have different inherent knowledge levels, and larger models have higher cognitive capacity, experiments with multiple model sizes might reveal more valuable insights for improving the social intelligence of LLMs.

ETHICS STATEMENT

All datasets used in this study's experiments (ToMi, HiToM, ExploreToM, ToMbench, OpenToM, SimpleToM, ToMATO, etc.) are publicly available benchmark datasets, and all large language models (LLMs) evaluated in this paper are publicly accessible and used through their official websites. This paper includes human performance baselines, but these serve solely as a reference for model performance with no other purposes. The paper does not mention any personally identifiable, sensitive, or proprietary information. Our reinforcement learning approach aims to improve LLMs' social intelligence performance, limited to method exploration and model evaluation. The authors declare no conflicts of interest with this submission. To our knowledge, this research complies with ICLR's ethical standards and presents no foreseeable ethical concerns.

REPRODUCIBILITY STATEMENT

We have made extensive efforts to ensure the reproducibility of our work. The datasets used for In-Domain training, as well as those used for In-Domain and Out-of-Domain evaluation, are explicitly described in Section 2. For the methodology, we provide complete reward design and implementation procedure. Training details, including hyperparameter settings, GPU devices, and other parameters, are thoroughly documented in the paper. In supplementary materials, we also submit the source code and datasets required to reproduce our method. Upon acceptance, we will release data, models, and training and evaluation code to facilitate post-training research on social intelligence in LLMs.

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

# A APPENDIX

## A.1 PRELIMINARY EXPERIMENTS——DEEPSEEK-R1'S EVALUATION PERFORMANCE ON TOMBENCH

We evaluate the performance of the DeepSeek-R1 model on ToMBench, and compare it with models from the GPT-4 (OpenAI, 2023) series, Claude series (Anthropic, 2024), and Qwen-Max (Cloud, 2023) models. The comparison results are shown in Table 5.

## A.2 RELATED WORK

**Strategies for Enhancing LLMs' Cognitive Development in Social Domain**    To enhance LLMs' cognitive development in the social domain, existing methods can be mainly divided into three categories: (1) Prompt-based Methods (2) Tool-based Methods (3) Model-based Methods. SimToM (Wilf et al., 2024) prompts LLMs to adopt perspective-taking cognitive strategies, while PercepToM (Jung et al., 2024) improves perception-to-belief inference by extracting relevant contextual details. Meanwhile, Huang et al. (2024a) utilizes an LLM as a world model to track changes in environmental entity states and character belief states. Hou et al. (2024b) proposes a belief solver that transforms higher-order social cognition problems into lower-order social cognition problems based on intersections over time sets, while SymbolicToM (Sclar et al., 2023) uses graphical representations to track characters' beliefs. Additionally, AutoToM, MMToM, and MuMA-ToM (Zhang et al., 2025; Shi et al., 2025; Jin et al., 2024) propose Bayesian model-based methods. However, **there remains a notable research gap in systematic exploration from post-training and test-time intervention perspectives.**

**LLM Post-Training**    Supervised Fine-Tuning (SFT) and Reinforcement Learning (RL) have been widely used in LLM Post-Training to improve performance on specific tasks. There are also multiple studies exploring and analyzing these two different Post-Training methods, such as "What LLMs Can—and Still Can't—Solve after SFT?" (Hou et al., 2024a; Chen et al., 2025; Sun et al., 2025) and "SFT Memorizes, RL Generalizes" observations (Chu et al., 2025). Rule-based RL has already been widely applied to multiple domains beyond mathematics (Hu et al., 2025) and coding (Wei et al., 2025), such as image classification, emotion classification tasks (Li et al., 2025b; He et al., 2025), search engine calling (Jin et al., 2025; Song et al., 2025), video reasoning (Feng et al., 2025a), logic puzzles (Xie et al., 2025), machine translation (Feng et al., 2025b), and more (Li et al., 2025a; Xia & Luo, 2025; Zhou et al., 2025).

**Test-Time Scaling**    Test-time scaling methods can be divided into 1) Sequential, where later computations depend on earlier ones (e.g., a long reasoning trace), and 2) Parallel, which relies on multiple solution attempts generated in parallel and selecting the best via majority voting or reward model (process-based or outcome-based) (Snell et al., 2024; Brown et al., 2024; Liu et al., 2024; Huang et al., 2024b; Wang et al., 2024; Zeng et al., 2024; Qi et al., 2024). s1 in the mathematics domain, m1 (Huang et al., 2025) in the medical domain, and Z1 (Yu et al., 2025) in the code domain are all recent research works related to test-time scaling (Muennighoff et al., 2025). The "budget forcing" proposed in s1 refers to appending a "Wait" token to the model's current reasoning trace to encourage the model to engage in more thinking and exploration. In m1, it is precisely about applying budget forcing to the medical domain.

## A.3 TRAINING PARAMETER CONFIGURATIONS

**SFT-based Methods.**    The SFT training process employs full-parameter fine-tuning with DeepSpeed ZeRO-3 optimization (Rajbhandari et al., 2020). We conduct three epochs of training in bfloat16 precision, with a learning rate of 5e-5, a per-device train batch size of 1, and a cutoff length of 16384. We save the model once every 500 training steps.

**RL-based Methods.**    The RL training process uses a train batch size of 8, with the maximum input prompt length set to 1536 and the maximum response length set to 2048. The learning rate is set to 3e-7, and the KL loss coefficient is set to 0.001 to ensure sufficient optimization of the policy model. The temperature coefficient is set to 1.0. In the GRPO algorithm, the number of samples in each

Table 5: DeepSeek-R1's evaluation performance on ToMBench, where UOT represents Unexpected Outcome Test, SIT represents Scalar Implicature Task, PST represents Persuasion Story Task, FBT represents False Belief Task, AST represents Ambiguous Story Task, HT represents Hinting Test, SST represents Strange Story Task, FRT represents Faux-pas Recognition Test.

| Model | UOT | SIT | PST | FBT | AST | HT | SST | FRT | AVG |
|---|---|---|---|---|---|---|---|---|---|
| GPT-4-0613 | 0.713 | 0.49 | 0.58 | 0.863 | 0.84 | 0.796 | 0.83 | 0.766 | 0.735 |
| GPT-4-1106 | 0.767 | 0.48 | 0.61 | 0.908 | 0.83 | 0.883 | 0.762 | 0.786 | 0.753 |
| Claude-3.5-Sonnet-20240620 | 0.733 | 0.505 | 0.61 | 0.858 | 0.895 | 0.971 | 0.771 | 0.834 | 0.772 |
| Claude-3.5-Sonnet-20241022 | 0.78 | 0.615 | 0.61 | 0.88 | 0.875 | 0.961 | 0.862 | 0.83 | 0.802 |
| Qwen-Max-0919 | 0.74 | 0.475 | 0.62 | 0.898 | 0.815 | 0.874 | 0.813 | 0.802 | 0.755 |
| DeepSeek-R1 | 0.797 | 0.565 | 0.56 | 0.895 | 0.845 | 0.951 | 0.848 | 0.809 | 0.784 |
| Human | **0.893** | **0.755** | **0.70** | **0.868** | **0.95** | **0.971** | **0.892** | **0.804** | 0.854 |

group is set to 8. The model is saved every 500 steps, and a validation evaluation is performed every 25 steps.

## A.4 THE USE OF LARGE LANGUAGE MODELS (LLMS)

In compliance with the ICLR 2026 disclosure requirements on language model usage, we confirm that the use of LLMs in this study was strictly limited to linguistic refinement. Specifically, they were employed to improve syntactic structure, enhance academic style, standardize terminology, and increase the readability of technical content, thereby facilitating clearer scientific communication. Importantly, LLMs were not involved in generating research ideas, designing methodologies, or contributing to scientific conclusions; these aspects were carried out solely by the authors.

## A.5 CASE STUDY

In the Figure 4, we present how LLM adaptively employs different cognitive modes to address various data types in the social domain. For question with interpersonal reasoning depth of 4, LLM adopts the cognitive mode of <think> + <answer> to solve. For a simple social situational cognition question, LLM adopts the cognitive mode of <answer> or <social context understanding> + <answer> to solve.

## A.6 SAMPLE EXAMPLES FROM DATA SOURCE

For the data sources we use, we present sample examples from Figure 5 to Figure 12. Figure 5 presents a sample example from the HiToM data source, where the Story consists of Event lines, with its notable characteristic being the inclusion of interpersonal reasoning questions with deeper reasoning depth. Figure 6 presents a sample example from the ExploreToM data source, whose Story includes some advanced socio-cognitive events, such as "told privately," "witnessed this action in secret," "got distracted," etc. Figures 7 and 8 present sample examples from the ToMBench and SocialIQA data sources, respectively, examining models' cognition of social situations, with the data format being (social situation, question, choices). Figure 9 presents a sample example from the SimpleToM data source, which not only assesses social-situation cognition but also behavior judgment. Figure 10 presents sample examples from the OpenToM data source, which includes Location questions and Attitude questions. Figures 11 and 12 present sample examples from the ToMATO data source with reasoning depths of 1 and 2, respectively. Using Sotopia-generated agent-agent dialogues, ToMATO evaluates the LLMs' ability to infer agents' mental states during dialogue interaction.

<answer> A: Knows </answer>

[Structure Validation]
Pattern 1: <answer> </answer>

<social context understanding> Qing Shi feels socially marginalized, finds it difficult to adjust to the new environment and the friendships. It is natural that new classmates are uncertain about how to approach or interact with him. </social context understanding> <answer> B: He feels curious, but also a bit nervous. </answer>

[Structure Validation]
Pattern 2: <social context understanding> </social context understanding> <answer> </answer>

<think> The question is asking about Abigail's belief about Jackson's belief about Amelia's belief about Hannah's belief regarding the location of the peas. From the story, we know:
Hannah moved the peas to the blue_bathtub (line 10).
Hannah exited the patio (line 11).
Jackson, Mila, Amelia, and Abigail entered the waiting_room (line 14).
Hannah, Amelia, and Mila entered the lounge (line 15).
Hannah moved the lemon to the red_envelope (line 17).
Hannah exited the lounge (line 18).
Amelia moved the lemon to the blue_container (line 19).
Amelia exited the lounge (line 20).
Jackson dislikes the beans (line 21).
Mila moved the lemon to the green_box (line 22).
Hannah likes the green_crate (line 23).
Hannah exited the lounge (line 24).
Hannah, Abigail, Mila, and Jackson entered the waiting_room (line 25).
Hannah, Abigail, Mila, and Jackson entered the closet (line 26).
The sweet_potato is in the red_bucket (line 27).
Hannah moved the sweet_potato to the green_cupboard (line 28).
Hannah exited the closet (line 29).
Abigail made no movements and stayed in the closet for 1 minute (line 30).
Abigail exited the closet (line 31).
Mila made no movements and stayed in the closet for 1 minute (line 32).
Mila exited the closet (line 33).
Jackson made no movements and stayed in the closet for 1 minute (line 34).
Jackson exited the closet (line 35).
Hannah, Abigail, Mila, and Jackson entered the waiting_room (line 36).

Since the question is about the location of the peas, and Hannah moved the peas to the blue_bathtub, and no one else moved the peas after that, the location of the peas is the blue_bathtub. </think>
<answer> L. green_basket </answer>

[Structure Validation]
Pattern 3: <think> </think> <answer> </answer>

Figure 4: LLM adaptively employs different cognitive modes to address social situation cognition and interpersonal reasoning question in the social domain.

*Story:*
*{Chapter 1}*
[Liam, *Jack*, *Hannah*, *Owen* and *Noah* entered the pantry]. [The eggplant is in the green_envelope]. [*Liam* made no movements and stayed in the pantry for 1 minute]. [*Liam* exited the pantry]. [*Noah* lost his gloves]. [*Jack* moved the eggplant to the green_bathtub]. [*Jack* exited the pantry]. [*Hannah* moved the eggplant to the blue_cupboard]. [*Hannah* exited the pantry]. [*Owen* moved the eggplant to the green_basket]. [*Owen* exited the pantry]. [*Noah* moved the eggplant to the green_envelope]. [*Noah* exited the pantry].

*{Chapter 2}*
*Liam*, *Jack*, *Hannah*, *Owen* and *Noah* entered the waiting_room
*{Action List}*

*{Chapter 3}*
*Liam*, *Jack*, *Hannah*, *Owen* and *Noah* entered the dining_room
*{Action List}*

*Question:* Where does *Owen* think *Jack* thinks *Hannah* thinks *Noah* thinks the eggplant is?

*Choices:* A. green_treasure_chest, B. blue_treasure_chest, C. red_crate, D. blue_drawer, E. green_pantry, F. green_envelope, G. green_bathtub, H. blue_cupboard, I. blue_container, J. green_basket, K. blue_bottle, L. green_drawer, M. red_box, N. red_pantry, O. red_envelope

Figure 5: Sample example from HiToM. Data format: (Social-Event Lines, Question, Choices).

*Story:*
*Sophia* *told privately* to *Bryce* that she is in the art studio. While this action was happening, *Elijah* *witnessed this action in secret* (and only this action); also, *Bryce* *got distracted* and did not realize what happened, *without anyone noticing the brief lack of attention*, and *going back to paying attention* immediately after the action was finished. *Sophia* entered the multipurpose room. *Sophia* entered the art studio. *Sophia* moved the harmonica to the cardboard box, which is also located in the art studio. *Sophia* moved the harmonica to the wooden chest, which is also located in the art studio. *Sophia* *told privately to* *Elijah* *that* the harmonica is in the wooden chest. *Bryce* entered the multipurpose room. *Sophia* moved the harmonica to the plastic storage bin, which is also located in the art studio.

*Question:* In which container does *Elijah* think that *Sophia* will search for the harmonica?

Figure 6: Sample example from ExploreToM. Data format: (Social-Event Lines, Question).

**Social Situation**: Xiao Li and Xiao Zhang discuss weekend plans in the dormitory. Xiao Li suggests: "We can go to the amusement park on the weekend, it should be fun." Xiao Zhang replies: "Hmm, that works." Xiao Li curiously asks: "What happens to you recently?" Xiao Zhang softly says: "Nothing, just sometimes I miss home." Xiao Li casually laughs: "Miss home? We are so busy with college life, where is the time to miss home!" Xiao Zhang hears this and remains silent, does not respond.",

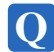 If there is, which sentence is inappropriate?

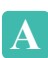 (A) We can go to the amusement park on the weekend, it should be fun.
(B) Nothing, just sometimes I miss home.
(C) Miss home? We are so busy with college life, where is the time to miss home!
(D) There is no inappropriate sentence

**Social Situation**: Aunt Yang slips and injures herself in the heavy snow. Her neighbor, Uncle Wang, helps her in time and takes her to the hospital. Uncle Wang also takes care of her house and pets

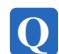 What will Mary likely do next?

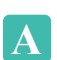 (A) Angry
(B) Ashamed
(C) Curious
(D) Grateful

Figure 7: Sample example from ToMbench. Data format: (Social Situation, Question, Choices).

### REASONING ABOUT MOTIVATION

The bag of potato chips has moldy chips in it. Mary picks up the bag in the supermarket and walks to the cashier.

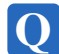 Is Mary likely to be aware that "The bag of potato chips has moldy chips in it."?

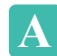 (A) Yes.
(B) No

### REASONING ABOUT MOTIVATION

The bag of potato chips has moldy chips in it. Mary picks up the bag in the supermarket and walks to the cashier.

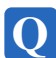 What will Mary likely do next?

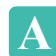 (A) Pay for the chips.
(B) Report the moldy chips

Figure 8: Sample example from SimpleToM, SocialIQA. Data format: (Social Situation, Question, Choices).

### A.7 COMPREHENSIVE COMPARISON OF OUR METHOD AND EXISTING LLMs

Through extensive literature research, we have collected as much data as possible on the performance of existing LLMs on corresponding In-Domain and Out-of-Domain data sources, and conduct a comprehensive comparison with the performance of our proposed TimeHC-RL method (applied to 7B models) as shown in Table 6 and 7.

*Narrative* : The bag of potato chips has moldy chips in it. Mary picks up the bag in the supermarket and walks to the cashier.
*Question* : Is Mary likely to be aware that "The bag of potato chips has moldy chips in it."?
*Choices* : (A) Yes. (B) No
**Mental State**

*Narrative* : The bag of potato chips has moldy chips in it. Mary picks up the bag in the supermarket and walks to the cashier.
*Question* : What will Mary likely do next?
*Choices* : (A) pay for the chips. (B) report the moldy chips
**Behavior Prediction**

Figure 9: Sample example from SimpleToM, SocialIQA. Data format: (Social Situation, Question, Choices).

*Narrative* : Diego and Amir were both residents of the same apartment complex. They had known each other for quite some time, but they couldn't be more different in their tastes and preferences. One thing that particularly divided them was their opinion on scarves. Diego despised scarves, finding them to be unnecessary and bothersome. On the other hand, Amir adored scarves, always wearing one to complete his outfit.

One sunny afternoon, both Diego and Amir happened to stroll into the patio at the same time. As they approached the central basket, their eyes fell upon a colorful scarf lying inside. Diego's face contorted in disdain while Amir's eyes lit up with delight.

In that moment, without exchanging any words, Diego swiftly reached into the basket and snatched the scarf. Amir watched curiously as Diego took a few steps towards a nearby donation bin. With a resolute expression, Diego dropped the scarf into the bin, relieving himself of its presence.

And just like that, the scarf that once rested in the patio basket had found a new temporary home in the donation bin, waiting to be discovered by someone who would appreciate its warmth and beauty. Diego turned around to leave the patio, completely unaware that his actions had not gone unnoticed by Amir.

*Question* : As Amir, what is your attitude towards Diego's action of moving scarf from basket to a donation bin?

**Attitude Question**

---

*Narrative* : Diego and Amir were both residents of the same apartment complex. They had known each other for quite some time, but they couldn't be more different in their tastes and preferences. One thing that particularly divided them was their opinion on scarves. Diego despised scarves, finding them to be unnecessary and bothersome. On the other hand, Amir adored scarves, always wearing one to complete his outfit.

One sunny afternoon, both Diego and Amir happened to stroll into the patio at the same time. As they approached the central basket, their eyes fell upon a colorful scarf lying inside. Diego's face contorted in disdain while Amir's eyes lit up with delight.

In that moment, without exchanging any words, Diego swiftly reached into the basket and snatched the scarf. Amir watched curiously as Diego took a few steps towards a nearby donation bin. With a resolute expression, Diego dropped the scarf into the bin, relieving himself of its presence.

And just like that, the scarf that once rested in the patio basket had found a new temporary home in the donation bin, waiting to be discovered by someone who would appreciate its warmth and beauty. Diego turned around to leave the patio, completely unaware that his actions had not gone unnoticed by Amir.

*Question* : From Diego's perspective, is the scarf in its initial location by the end of the story?

**Location Question — First Order**

---

*Narrative* : Diego and Amir were both residents of the same apartment complex. They had known each other for quite some time, but they couldn't be more different in their tastes and preferences. One thing that particularly divided them was their opinion on scarves. Diego despised scarves, finding them to be unnecessary and bothersome. On the other hand, Amir adored scarves, always wearing one to complete his outfit.

One sunny afternoon, both Diego and Amir happened to stroll into the patio at the same time. As they approached the central basket, their eyes fell upon a colorful scarf lying inside. Diego's face contorted in disdain while Amir's eyes lit up with delight.

In that moment, without exchanging any words, Diego swiftly reached into the basket and snatched the scarf. Amir watched curiously as Diego took a few steps towards a nearby donation bin. With a resolute expression, Diego dropped the scarf into the bin, relieving himself of its presence.

And just like that, the scarf that once rested in the patio basket had found a new temporary home in the donation bin, waiting to be discovered by someone who would appreciate its warmth and beauty. Diego turned around to leave the patio, completely unaware that his actions had not gone unnoticed by Amir.

*Question* : From Diego's perspective, does Amir think that the scarf is in its initial location by the end of the story?

**Location Question — Second Order**

Figure 10: Sample example from OpenToM. Data format: (Social Situation, Question, Choices).

## First Order

### Question

How does Liam feel when he says "Ethan, I've noticed you've been a bit distant lately, is everything okay?" ?

### Sotopia Agent

A

B

### Conversation

**Liam Johnson:** "Hi!"

**Ethan Johnson:** "Hi, how are you?"

**Liam Johnson:** "Ethan, I've noticed you've been a bit distant lately, is everything okay?"

**Ethan Johnson:** "Yeah, everything's fine, just been really busy with the restaurant and some personal stuff, you know how it is."

**Liam Johnson:** "Come on, Ethan, we've been friends for years, I can tell when something's bothering you. What's really going on?"

**Ethan Johnson:** "Alright, alright... to be honest, I've been having some trouble making ends meet, the restaurant's been slow and I'm falling behind on some bills, but I'll figure it out, I always do."

**Liam Johnson:** "Ethan, you're not going to 'figure it out' on your own, you need a plan and some guidance. Let me take a look at your finances and we can go over some options together, okay?"

**Ethan Johnson:** "Man, I appreciate that, Liam, but I don't want to be a burden, I can handle it, I just need to get my head out of my... okay, fine, yeah, maybe a fresh set of eyes would help, thanks."

**Liam Johnson:** "Alright, let's get down to business then. I'll take a look at your books and we'll get a plan in place. But next time, don't be so stubborn and come to me sooner, got it?"

**Ethan Johnson:** "Ha! Yeah, yeah, got it, I'll try to work on that, thanks, Liam. And thanks again, man, I owe you one... or a few thousand, it seems."

**Liam Johnson:** "Yeah, yeah, just pay me back in free meals at the restaurant, and we'll call it square. But seriously, let's take a close look at your expenses and see where we can cut back, okay?"

**Ethan Johnson:** "Aww, man, you're a lifesaver, Liam. Yeah, let's dig in and see where I can tighten the belt, I'm sure there's some fat to trim. And don't worry, you'll be eating like a king at my place for a long time to come."

**Liam Johnson:** "Alright, let's get to work then. I'll start going over the numbers and we can discuss some options. And don't make promises you can't keep, just focus on getting back on your feet, okay?"

**Ethan Johnson:** "Yeah, yeah, got it, thanks for keeping me grounded, Liam. I'll let you do your magic with the numbers, and I'll focus on getting the restaurant back on track. You're right, one step at a time."

**Liam Johnson:** "That's the spirit, Ethan. Just remember, I'm not a miracle worker, you need to put in the effort too. Let's schedule a follow-up meeting for next week and we'll reassess where we're at, sound good?"

**Ethan Johnson:** "Absolutely, sounds like a plan, Liam. I'll make sure to have all my ducks in a row by then. And thanks again, man, just knowing you've got my back makes me feel a hell of a lot better.

Figure 11: Sample example from ToMATO (First order). Data format: (Conversation, Question, Choices).

## Second Order

### Question

When Ethan says "Yeah, everything's fine, just been really busy with the restaurant and some personal stuff, you know how it is.", how does he think that Liam feels?

### Sotopia Agent

A

B

### Conversation

**Liam Johnson:** "Hi!"

**Ethan Johnson:** "Hi, how are you?"

**Liam Johnson:** "Ethan, I've noticed you've been a bit distant lately, is everything okay?"

**Ethan Johnson:** "Yeah, everything's fine, just been really busy with the restaurant and some personal stuff, you know how it is."

**Liam Johnson:** "Come on, Ethan, we've been friends for years, I can tell when something's bothering you. What's really going on?"

**Ethan Johnson:** "Alright, alright... to be honest, I've been having some trouble making ends meet, the restaurant's been slow and I'm falling behind on some bills, but I'll figure it out, I always do."

**Liam Johnson:** "Ethan, you're not going to 'figure it out' on your own, you need a plan and some guidance. Let me take a look at your finances and we can go over some options together, okay?"

**Ethan Johnson:** "Man, I appreciate that, Liam, but I don't want to be a burden, I can handle it, I just need to get my head out of my... okay, fine, yeah, maybe a fresh set of eyes would help, thanks."

**Liam Johnson:** "Alright, let's get down to business then. I'll take a look at your books and we'll get a plan in place. But next time, don't be so stubborn and come to me sooner, got it?"

**Ethan Johnson:** "Ha! Yeah, yeah, got it, I'll try to work on that, thanks, Liam. And thanks again, man, I owe you one... or a few thousand, it seems."

**Liam Johnson:** "Yeah, yeah, just pay me back in free meals at the restaurant, and we'll call it square. But seriously, let's take a close look at your expenses and see where we can cut back, okay?"

**Ethan Johnson:** "Aww, man, you're a lifesaver, Liam. Yeah, let's dig in and see where I can tighten the belt, I'm sure there's some fat to trim. And don't worry, you'll be eating like a king at my place for a long time to come."

**Liam Johnson:** "Alright, let's get to work then. I'll start going over the numbers and we can discuss some options. And don't make promises you can't keep, just focus on getting back on your feet, okay?"

**Ethan Johnson:** "Yeah, yeah, got it, thanks for keeping me grounded, Liam. I'll let you do your magic with the numbers, and I'll focus on getting the restaurant back on track. You're right, one step at a time."

**Liam Johnson:** "That's the spirit, Ethan. Just remember, I'm not a miracle worker, you need to put in the effort too. Let's schedule a follow-up meeting for next week and we'll reassess where we're at, sound good?"

**Ethan Johnson:** "Absolutely, sounds like a plan, Liam. I'll make sure to have all my ducks in a row by then. And thanks again, man, just knowing you've got my back makes me feel a hell of a lot better."

Figure 12: Sample example from ToMATO (Second order). Data format: (Conversation, Question, Choices).

Table 6: A comprehensive performance comparison of our method and existing LLMs on several In-Domain data sources.

| Model | ToMi | ExploreToM | ToMBench | SocialIQA | HiToM (Third) | HiToM (Fourth) |
|---|---|---|---|---|---|---|
| ChatGLM3-6B | - | - | 0.43 | - | - | - |
| Llama2-13B-Chat | - | - | 0.43 | - | - | - |
| Llama3.1-8B-Instruct | 0.68 | - | - | - | - | - |
| Llama3.1-70B-Instruct | - | 0.48 | - | - | - | - |
| Baichuan2-13B-Chat | - | - | 0.49 | - | - | - |
| Guanaco-65B | - | - | - | - | 0.08 | 0.06 |
| Claude-instant | - | - | - | - | 0.08 | 0.07 |
| Mistral-7B | - | - | 0.49 | - | - | - |
| Mixtral-8x7B | - | - | 0.54 | - | - | - |
| Mixtral-8x7B-Instruct | - | 0.47 | - | - | - | - |
| Qwen2.5-7B-Instruct | 0.67 | - | - | - | - | - |
| Qwen-14B-Chat | - | - | 0.58 | - | - | - |
| GPT-3.5-Turbo | - | - | - | - | 0.03 | 0.01 |
| GPT-3.5-Turbo-0613 | - | - | 0.58 | - | - | - |
| GPT-3.5-Turbo-1106 | - | - | 0.59 | - | - | - |
| GPT-4-32K | - | - | - | - | 0.18 | 0.15 |
| GPT-4-0613 | - | - | 0.71 | - | - | - |
| GPT-4-1106 | - | - | 0.74 | - | - | - |
| GPT-4o | - | 0.47 | - | - | - | - |
| TimeHC-RL (7B) | **0.96** | **0.95** | **0.82** | **0.78** | **0.68** | **0.64** |

Table 7: A comprehensive performance comparison of our method and existing LLMs on several Out-of-Domain data sources.

| Model | ToMATO(First) | ToMATO(Second) | SimpleToM(Behavior) | OpenToM(Attitude) | OpenToM(Location) |
|---|---|---|---|---|---|
| Llama2-Chat-7B | - | - | - | 0.24 | 0.37 |
| Llama2-Chat-13B | - | - | - | 0.37 | 0.37 |
| Llama2-Chat-70B | - | - | - | 0.41 | 0.34 |
| Llama3-8B | 0.54 | 0.40 | - | - | - |
| Llama3-70B | 0.81 | 0.71 | - | - | - |
| Llama3.1-8B | 0.64 | 0.46 | **0.54** | - | - |
| Llama3.1-70B | **0.82** | 0.73 | - | - | - |
| Llama3.1-405B | - | - | 0.10 | - | - |
| Gemma2 | 0.79 | 0.71 | - | - | - |
| Claude-3-Kaiku | - | - | 0.17 | - | - |
| Claude-3-Opus | - | - | 0.10 | - | - |
| Claude-3.5-Sonnet | - | - | 0.25 | - | - |
| Mistral-7B | 0.65 | 0.56 | - | - | - |
| Mixtral-8x7B | 0.65 | 0.57 | - | - | - |
| Mixtral-8x7B-Instruct | - | - | - | 0.40 | 0.47 |
| GPT-3.5 | - | - | 0.29 | - | - |
| GPT-3.5-Turbo | 0.60 | 0.51 | - | 0.38 | 0.41 |
| GPT-4 | - | - | 0.20 | - | - |
| GPT-4-Turbo | - | - | - | 0.54 | 0.54 |
| GPT-4o-mini | 0.77 | 0.69 | - | - | - |
| o1-mini | - | - | 0.27 | - | - |
| TimeHC-RL (7B) | 0.80 | **0.80** | 0.35 | **0.60** | **0.73** |

