# OpenReview forum: "TimeHC-RL: Temporal‑aware Hierarchical Cognitive Reinforcement Learning for Enhancing LLMs' Social Intelligence"
_ICLR.cc/2026/Conference — ICLR 2026 Conference Withdrawn Submission_

### Official Review · Reviewer_wECB · 2025-10-25

**Soundness:** 2
**Presentation:** 3
**Contribution:** 2
**Rating:** 2
**Confidence:** 4

**Summary:**

The paper proposes a new reinforcement learning (RL) framework, TimeHC-RL, to improve the social intelligence of LLMs. It combines temporal awareness with a dual-system cognitive structure inspired by psychological theories (System 1: intuition; System 2: deliberation). The approach introduces a hierarchical cognition framework and a temporal advantage-level reward, which encourages models to reason coherently across temporally ordered social events. Experiments across eight Theory of Mind (ToM) datasets (e.g., ToMBench, HiToM, SocialIQA) compare TimeHC-RL with several baselines, including SFT, long-thought SFT, and rule-based RL. Results show that TimeHC-RL improves both in-domain and out-of-domain performance, achieving parity with larger models such as DeepSeek-R1 and OpenAI-O3.

**Strengths:**

- experimental coverage: Evaluation on eight datasets (ToMi, HiToM, ToMBench, SocialIQA, etc.) provides a rich and detailed overview of the model's behaviour, including both in-domain and out-of-domain tests.

- Integration of cognitive perspectives: The use of Dual-System Theory (System 1/System 2) gives the work an interesting interdisciplinary dimension between cognitive psychology and reinforcement learning.

- Introduction of a nice reward: The temporal advantage-level reward represents a novel idea for modelling the temporal coherence of social events, an aspect rarely considered in RL approaches.

**Weaknesses:**

- Overly ambitious and conceptually diffuse. The work merges cognitive psychology (dual-process theory), temporal reasoning, and reinforcement learning, yet fails to sufficiently ground its interactions in formal or mechanistic terms. The psychological framing often feels ornamental rather than integral to the method.

- Figures and tables are complicated to read. Many are dense, overcrowded, and almost illegible, with tiny fonts, unclear legends, and excessive textual overlays. This severely undermines the clarity and interpretability of the results. The authors must redesign the figures entirely, simplifying visual presentation and highlighting only the most relevant comparisons.

- Lack of ablation or causal evidence. The performance gains attributed to the temporal and hierarchical components are not convincingly disentangled. No ablation isolates the effects of temporal rewards, hierarchical structure, or training data diversity.

- Empirical overreach. The claim that the 7B model “rivals” DeepSeek-R1 and OpenAI-O3 is overstated, given the modest numerical margins and lack of significance testing.

**Questions:**

- How can you demonstrate that TimeHC-RL’s gains derive from temporal or hierarchical mechanisms rather than dataset scale or reward shaping?

- In what sense does your dual-system framing extend beyond metaphor? Does it correspond to measurable differences in reasoning dynamics?

---

### Official Review · Reviewer_xiHh · 2025-11-01

**Soundness:** 2
**Presentation:** 2
**Contribution:** 2
**Rating:** 2
**Confidence:** 4

**Summary:**

This work proposes TimeHC-RL, a reinforcement learning framework built upon Group Relative Policy Optimization, which introduces two key components: (1) a hierarchical cognitive template to guide the model in adaptively selecting reasoning modes during response generation, and (2) a temporal advantage-level reward that encourages the model to construct temporal logic by contrasting responses based on correctly ordered versus shuffled event sequences. The authors conduct systematic evaluations across multiple datasets and report that a 7B backbone model trained with TimeHC-RL achieves improvements on both in-domain and out-of-domain tasks, matching or even surpassing the performance of larger or more advanced baselines.

**Strengths:**

1. Exploring the social intelligence of large language models is an interesting research direction.

2. Experiments are conducted across multiple datasets, with both in-domain and out-of-domain settings.

3. The performance gain on out-of-domain setting is more pronounced.

**Weaknesses:**

1. The Introduction lacks clarity and logical flow, making it difficult for readers to quickly grasp the core contribution. Specifically:
    * The distinction between “social domains” and “IQ-related domains” should be introduced early and explicitly, rather than requiring readers to infer it from a lengthy analysis (lines 48–91) and the beginning of Section 2 (line 145).
    * The discussion of DeepSeek-R1’s performance in social reasoning (lines 48–91) is overly verbose, with only lines 87–91 containing the key insight. Moreover, this analysis reads more like speculation than evidence-backed observation.
    * The contribution statement is fragmented, appearing in both lines 100–107 and 136–139, making it hard to identify the central innovation at a glance.

2. Do all questions have temporal logic dependencies? If not, what specific performance differences do the authors' methods have when the problem has no such dependency? A more detailed comparison and analysis is needed.

3. There is a lack of quantitative analysis regarding the hierarchical cognitive modes. For instance, what proportion of questions trigger each of the three reasoning modes across different datasets? And what are the respective accuracy rates for each mode?

4. The paper lacks direct comparisons with several relevant methods cited in the Related Work.

**Questions:**

1. In common GRPO setups, the Accuracy Reward typically assigns zero (or no penalty) for incorrect answers. However, this paper sets the reward to –1.5 for wrong responses. Is there a specific rationale behind this choice?

2. When the answer is correct, the Format Reward is set to 1, the Accuracy Reward to 2, but the proposed temporal advantage-level reward is only 0.1—significantly smaller in magnitude. Given this large disparity, why that such a minor reward component consistently yields a 2%–4% performance gain? Moreover, why was such a small value chosen, and how sensitive is the overall performance to variations in this reward scale?

3. The RL training configuration uses a maximum input length of 1536 and output length of 2048, which are well within the capacity of standard LLMs like the base Qwen model. Given this, what is the justification for using the 1M-context version of Qwen? Does the long-context capability actually get utilized in the experiments?

4. The paper only evaluates the method with Qwen as the backbone. I wander know whether similar improvements can be observed with other open-source LLMs (e.g., Llama-3 or Mistral).

---

### Official Review · Reviewer_c7ZX · 2025-11-01

**Soundness:** 3
**Presentation:** 2
**Contribution:** 2
**Rating:** 2
**Confidence:** 4

**Summary:**

This paper explores the enhancement of social intelligence in LLMs through a temporal-aware hierarchical cognitive reinforcement learning framework, TimeHC-RL. The idea of combining temporal dynamics with multi-level cognition is conceptually appealing and relevant to cognitive modeling. The experimental setup covers multiple datasets and post-training paradigms. However, the paper lacks ablation or interpretive analysis to clarify the independent effects of its components. The study is further limited by the use of a single backbone model, insufficient hyperparameter analysis, and reliance on multiple-choice evaluation, which may not fully reflect reasoning quality. Overall, further analysis and validation are needed to solidify the claimed contributions.

**Strengths:**

- This paper introduces TimeHC-RL, a temporal-aware hierarchical cognitive reinforcement learning framework. The idea of combining temporal dynamics with multi-level cognition is conceptually appealing to cognitive modeling in LLMs.
- The experimental setup spans multiple datasets and post-training paradigms with both in-domain and out-of-domain evaluations.

**Weaknesses:**

- Although TimeHC-RL shows overall performance improvement, the paper does not disentangle the independent contributions of the temporal reward and the hierarchical cognitive framework.
- Only one backbone model is trained, which may restrict the generalization ability of the proposed approach.
- The Temporal Advantage-level Reward depends on multiple hyperparameters (α, µ), but the paper does not analyze model performance under different parameter configurations.
- The paper mentions that majority voting and budget forcing show limited or no improvement, but it lacks in-depth analysis of the reasons behind their ineffectiveness.
- Since almost all evaluations are multiple-choice questions, the authors do not assess potential reasoning hacking, and there may be cases where the model answers correctly by guessing rather than through genuine reasoning.
- The improvement of TimeHC-RL over RL with System 1 and RL with System 2 appears modest. The paper would benefit from a deeper analysis of the underlying factors contributing to these gains.

**Questions:**

- While the authors select three cognitive modes: intuition, surface-level thinking, and deliberate thinking through a prompting template, it remains unclear what the actual distinction is between surface-level thinking and deliberate thinking. These two modes are guided by different tags (for example, and ), but it is uncertain whether the model truly understands and differentiates these tags and exhibits genuinely distinct reasoning patterns.
- Have the authors compared TimeHC-RL with any stronger or more comparable baselines beyond Qwen2.5-7B-Instruct, GPT-4o, GPT-o3, DeepSeek-R1?

---

### Official Review · Reviewer_fCqR · 2025-11-05

**Soundness:** 3
**Presentation:** 2
**Contribution:** 3
**Rating:** 2
**Confidence:** 5

**Summary:**

This paper introduces TimeHC-RL (Temporal-aware Hierarchical Cognitive Reinforcement Learning), a novel method to significantly enhance the Social Intelligence of LLMs. TimeHC-RL uses a temporal-aware reward mechanism to explicitly enforce temporal reasoning, and employs a hierarchical cognition framework that guides the model to adaptively utilize different cognitive modes (System 1 to System 2) for social problem-solving. Experimental results show TimeHC-RL's superiority over existing RL methods, allowing a 7B model to match the performance of much larger, state-of-the-art models in complex social domain tasks.

**Strengths:**

1. Significant Advancement in Social Intelligence: It dramatically improves LLMs' performance in complex social reasoning tasks, enabling a 7B model (TimeHC-RL (7B)) to achieve comparable results to much larger SOTA models like DeepSeek-R1 and OpenAI-03.

2. Novel Temporal-Aware Mechanism: The introduction of a Temporal-aware Reward Mechanism is key, as it explicitly trains the model to respect and construct the logical timeline of social events and dialogues, a crucial aspect often overlooked by existing RL methods.

3. Adaptive Hierarchical Cognition: It proposes a Hierarchical Cognition Framework that allows the LLM to adaptively switch between different cognitive modes, from System 1 (Intuitive Reaction) to System 2 (Deep Deliberation), mimicking human flexible decision-making.

4. Valuable Scientific Insights: The study provides important findings, such as the differentiation between SFT's tendency towards memorization versus RL's tendency towards generalization, and demonstrates RL's effectiveness for interpersonal reasoning depth extrapolation.

**Weaknesses:**

1. The authors only use textual datasets, which limits the application scenarios. Other modality data also plays significant roles in social reasoning.
2. It will be better if the authors can use some figures to illustrate their methods.
3. Overclaiming the Term "Hierarchical": Weak Hierarchy: The framework essentially introduces three different prompting styles (System 1, System 1.5/Shallow, System 2/Deliberate) and trains the model to select one via self-prompting. This is better described as Adaptive Cognitive Mode Selection rather than a true hierarchy. And the authors implement this idea by simply prompting LLMs, which is too simple and not solid. These are soft constraints based on text generation, not hard architectural constraints. The model might still generate the required tags without performing the actual deep reasoning, leading to tag-spoofing or superficial compliance.
Lack of Structural Recursion: In standard Hierarchical Reinforcement Learning (HRL), the hierarchy involves a nested structure where higher-level policies set subgoals for lower-level policies. TimeHC-RL's structure is a flat selection between different output formats, lacking this deep, recursive structural control.
4. The Why is Unclear: The mechanism only guides what the model should output (a thought process or a direct answer), but the underlying process of how the model decides which mode is truly optimal (the metacognitive control) is still hidden within the LLM's black box and remains difficult to audit.
5. Data Dependency: The model is trained to select the cognitive mode based on the specific textual datasets used. There is a risk that its "adaptive selection" ability is simply memorizing cues from the training data rather than developing a robust, generalized metacognitive ability to judge true task complexity in novel, out-of-distribution (OOD) social scenarios.

**Questions:**

See above.

---

### Note · Authors · 2025-12-04

**Comment:**

We have decided to withdraw our submission. We sincerely thank the reviewers for their time and valuable opinions. We found their feedback to be very constructive and plan to incorporate their suggestions to make the paper more competitive for a future submission. Specifically, we will conduct more comprehensive and rigorous experiments on the RL component, including enriching the design of adaptive cognitive experiments, extensively exploring the hyperparameter settings, and systematically investigating the impact of reward functions. We thank the reviewers again for their insightful feedback.

**Withdrawal Confirmation:**

I have read and agree with the venue's withdrawal policy on behalf of myself and my co-authors.